# The Intersection of DNA Damage Response and Ferroptosis—A Rationale for Combination Therapeutics

**DOI:** 10.3390/biology9080187

**Published:** 2020-07-23

**Authors:** Po-Han Chen, Watson Hua-Sheng Tseng, Jen-Tsan Chi

**Affiliations:** 1Department of Molecular Genetics and Microbiology, Duke University School of Medicine, Durham, NC 27710, USA; po-han.chen.pc673@yale.edu (P.-H.C.); watsontseng@gm.ym.edu.tw (W.H.-S.T.); 2Center for Genomic and Computational Biology, Duke University School of Medicine, Durham, NC 27710, USA; 3Department of Molecular, Cellular, and Developmental Biology, Yale University, New Haven, CT 06511, USA; 4Institute of Clinical Medicine, National Yang-Ming University, Taipei 112, Taiwan

**Keywords:** ferroptosis, DNA damage, ATM, ATR, p53, MDM2, MDMX

## Abstract

Ferroptosis is a novel form of iron-dependent cell death characterized by lipid peroxidation. While the importance and disease relevance of ferroptosis are gaining recognition, much remains unknown about its interaction with other biological processes and pathways. Recently, several studies have identified intricate and complicated interplay between ferroptosis, ionizing radiation (IR), ATM (ataxia–telangiectasia mutated)/ATR (ATM and Rad3-related), and tumor suppressor p53, which signifies the participation of the DNA damage response (DDR) in iron-related cell death. DDR is an evolutionarily conserved response triggered by various DNA insults to attenuate proliferation, enable DNA repairs, and dispose of cells with damaged DNA to maintain genome integrity. Deficiency in proper DDR in many genetic disorders or tumors also highlights the importance of this pathway. In this review, we will focus on the biological crosstalk between DDR and ferroptosis, which is mediated mostly via noncanonical mechanisms. For clinical applications, we also discuss the potential of combining ionizing radiation and ferroptosis-inducers for synergistic effects. At last, various ATM/ATR inhibitors under clinical development may protect ferroptosis and treat many ferroptosis-related diseases to prevent cell death, delay disease progression, and improve clinical outcomes.

## 1. Ferroptosis—Biological Processes and Genetic Determinants

Programmed cell death (PCD) plays a critical role in tissue homeostasis and many pathological conditions. Recently, ferroptosis was identified as a new form of PCD, with distinct biochemical, morphological, and cellular features [1]. Ferroptosis is characterized by iron-dependency, extensive lipid peroxidation, and plasma membrane damage [1,2]. Ferroptosis was first discovered by the mechanistic investigation of erastin-induced cell death [3]. Since then, significant progress has been made in the understanding of the biological processes of ferroptotic death and genetic determinants that promote or limit ferroptosis [1,2]. These genetic determinants typically affect lipid peroxidation and protection mechanisms by regulating the levels of cysteine, glutathione, iron, or lipids. 

Most mammalian cells have developed systems to neutralize lipid peroxidation and prevent ferroptosis. Genetic and chemical studies have pinpointed glutathione peroxidase (GPX4) as the critical regulator of this form of cell death [4]. Therefore, various GPX4 inhibitors can directly trigger ferroptosis [3]. In addition, because GPX4 relies on glutathione (GSH) and NADPH (dihydronicotinamide adenine dinucleotide phosphate) as cofactors, the depletion of GSH or NADPH can interfere with GPX4 function and induce ferroptosis. 

Cysteine is the limiting component of GSH that is imported via xCT from extracellular space in the form of cystine (a dimeric form of cysteine). xCT (encoded by *SLC7A11*) is the transmembrane antitransporter that mediates the cystine import through the export of glutamate. Therefore, xCT inhibitors (e.g., erastin or sulfasalazine) would deplete cysteine and GSH and trigger ferroptosis [3]. Similarly, when each amino acid is removed in several nutria-genetic screens, cystine emerged as indispensable for multiple cancer types, including triple-negative breast cancer [5], renal cell carcinoma [6], and nonsmall cell lung cell carcinoma [7].

NADPH is an essential cofactor for redox balance, and it participates in GSH regeneration. Therefore, the levels of NADPH in various cancer cells also predict ferroptosis sensitivity [8]. This idea is further corroborated by our recent finding of MESH1 as the first mammalian NADPH phosphatase, which is induced during ferroptosis to deplete NADPH [9]. Additionally, nuclear factor erythroid 2-related factor 2 (NRF2), the master transcriptional regulator of antioxidant stresses, can protect ferroptosis by inducing various components for GSH synthesis [10,11].

Other than GPX4, cell density and contact have recently been found to be robust determinants of ferroptosis in several different cancer cell lines via regulation of the Hippo effectors, YAP or TAZ, linking various mechanical stimuli, environmental cues, and metabolic status to ferroptosis [12,13,14,15]. Although GPX4 is crucial for protecting against ferroptosis, two studies have identified another independent system composed of FSP1–CoQ_10_ [16,17]. The myristoylation of FSP1 (ferroptosis-suppressing protein 1) allows its movement to the plasma membrane. FSP1 catalyzes the formation of reduced coenzyme Q10 (CoQ_10_), which serves as a radical-trapping antioxidant to neutralize lipid peroxidation and reactive oxygen species (ROS) and peroxidation. This novel mechanism of ferroptosis protection may provide additional insights and therapeutic opportunities. 

As the name implied, ferroptosis is an iron-dependent process since iron chelators robustly block ferroptosis [3]. Mammalian cells with high iron contents, such as erythrocyte-ingested macrophages [18] and hemochromatosis hepatocytes [19], are highly sensitive to ferroptosis. Iron has been speculated to drive the Fenton reaction and generate hydrogen peroxide to cause lipid peroxidation [20]. Regardless, ferroptosis can also be regulated by the levels and activities of these iron-related proteins. However, the role of iron in ferroptosis remains poorly understood.

In this review, we will focus on the recent discovery and therapeutic potential of novel ferroptosis regulators in the DNA damage response. The importance of the DDR pathway is shown by the fact that the germline or somatic mutations in DDR are the drivers of oncogenesis of many cancer types. Moreover, ionizing radiation is one pillar of cancer treatment that triggers DDR. Therefore, the role of DDR2 to promote ferroptosis may offer the rationale for synergistic therapeutic potential. Therefore, those new intersections between ferroptosis and DDR may be highly relevant during the initiation, progression, metastasis, and treatment response of different human cancers. 

## 2. Canonical Functions of DNA Damage Responses (DDRs)—The Role of ATR, ATM, p53, and MDM2

The integrity of the genome is essential for the faithful transfer of genetic materials to the progeny. However, certain endogenous levels of mutational changes are also crucial to enable genetic variations, adaptation, and evolution. Genomic DNA can be damaged from multiple exogenous and endogenous sources. The types of damage include replication errors, chemical-induced adducts, and crosslinks, ultraviolet light (UV)-induced injury, and single-strand (SSB) or double-strand (DSB) breaks created by ionizing radiation or chemical reactions [21,22]. To cope with these types of DNA damage and maintain genome integrity, DDR has been developed in all organisms to detect and sense DNA damage, transmit the signals to the appropriate effectors, and repair various forms of DNA damage and genomic insults. When DNA damage is not correctly repaired, DDR will trigger apoptosis and cell death programs to eliminate the unrepaired cells. In mammalian cells, the central sensors and upstream DDR kinases include the ATM (ataxia–telangiectasia mutated), ATR (ATM and Rad3-related), and DNA–PKs (DNA dependent protein kinase). These DDR kinases regulate the levels and activities of various effector proteins [23], including the guardian of genome integrity, p53, and its critical negative regulators, MDM2 (mouse double minute 2) and MDMX (murine double minute X).

### 2.1. ATM and ATR—The Kinases Sensing DNA Damages

ATM and ATR are two large serine/threonine kinases in the phosphatidylinositol-3-kinase-like kinase family (PIKK) [24,25]. ATM is mutated in ataxia–telangiectasia (AT; OMIM #208900) patients from all complementation groups, indicating that it is probably the single gene responsible for AT [26]. Ataxia–telangiectasia is a rare human autosomal recessive disorder. The affected AT individuals are immunodeficient, radiosensitive, and predisposed to the development of cancer [26]. Consistent with the role of ATM in DDR, AT cells have abnormal cell-cycle arrests and hypersensitivity to ionizing radiation. In contrast, ATR is identified based on its sequence homology to ATM, featuring its ability to prevent abnormal cell division or aneuploidy when activated during DNA damage [27].

Consistent with these genetic data, ATM and ATR are crucial for DDR and maintenance of genome stability. During DDR, ATM and ATR are activated within seconds and mediate the phosphorylation of hundreds of proteins at the Ser/Thr-Glu motifs [28,29,30]. Although structurally and functionally similar, ATM and ATR are triggered by distinct forms of DNA damage. The main trigger for ATM is DNA double-strand breaks. In contrast, ATR responds to a much broader spectrum of DNA damage, including single-strand DNA (ssDNA), stalled replication forks, and other replication stresses. In the absence of DNA damage, ATM exists as an inactive homodimer [31]. Upon DSB, ATM is recruited by the MRE11–RAD50–NBS1 complex (MRN complex) [32] on DSB lesion sites and becomes activated to phosphorylate its downstream effectors [33]. Downstream effectors of ATM are composed of hundreds of proteins, including Chk2, KAP1, RNF20, RNAP1/2, to mediate the p53 phosphorylation, epigenetic DNA repair, chromatin remodeling, and transcription inhibition, respectively [34,35]. Noteworthily, ATM itself can also activate p53 and its stability regulators, MDMX and MDM2 [36,37], to safeguard the genome from DSB damage.

On the other hand, the activation of ATR is a multistep process. Under replication stresses, ATR and its partner protein, ATRIP (ATR-interacting protein), are recruited by ssDNA-binding protein complexes and replication protein A (RPA) associated with the extended tracts of ssDNA [38,39]. Besides recruitment by RPA, ATR also interacts with TopBP1 and ETAA1 to achieve optimal activation and kinase activity [40,41]. Activated ATR phosphorylates a distinct set of downstream substrates, including Chk1 and CDC25A, to prevent premature mitosis and increase DNA repair for cell survival [42]. Although ATR and ATM kinases have some functional redundancies and interconnect with each other in the DDR pathways, they have distinct substrates and downstream functions. For example, ATR mainly activates the Fanconi anemia (FA) pathway to promote the repair of DNA interstrand crosslinks [43]. In contrast, ATM, but not ATR, regulates the development of the central nervous system, thus explaining the neurodegenerative and other neurological phenotypes in AT patients with defective ATM protein [44]. 

### 2.2. p53—The Guardian of the Mammalian Genome

One of the most critical targets of ATM/ATR in the DDR pathway is p53, often referred to as the guardian of the mammalian genome. ATM is the predominant upstream regulator of p53 through the regulation of Chk2 and MDM2/MDMx. While ATR is not required for p53 activation, ATR synergizes with p53 to ensure a successful replication checkpoint [45]. ATM/ATR defects lead to a significant delay in p53 upregulation in response to ionizing radiation or other types of DNA damage. p53 is one of the most critical tumor suppressor genes, and it is crucial for preventing cancer formation in vertebrates. As such, p53 mutations or polymorphisms are responsible for a significant portion of human cancer [46]. First, the inheritance of a p53 mutation leads to Li–Fraumeni syndrome (LFS, OMIM #151623), characterized by various early-onset cancers, such as breast, brain, and adrenal cancer and sarcomas. Secondly, somatic mutations in p53 are found in ~50% of all human cancers, including more than 90% of ovarian and uterine carcinoma. Additionally, the coding and noncoding regions of p53 contain at least hundreds of single nucleotide polymorphisms (SNPs; germline variants), which may alter p53 functions with significant effects on cancer susceptibility, progression, and response to various therapeutics [47]. Functionally, p53 is a transcription factor that elicits antitumorigenesis effects such as cell senescence, cell-cycle arrest, and cell death by affecting the expression of its target genes (e.g., p21, Puma, Bax) [48]. 

### 2.3. MDM2/MDMX—The Main Brake that Restrains p53

Given its essential role and dramatic effects on cellular phenotypes, p53 is elaborately regulated by MDM2/MDMX through several mechanisms [49]. In nonstressed cells, p53 is continuously translated but rapidly ubiquitinated by MDM2/MDMX and degraded by proteasome on the protein level [50,51,52,53]. MDM2 also inhibits p53 by other means, binding to the transactivation domain of p53 to curb its transcription activity [54,55]. Furthermore, MDM2 interferes with the nuclear localization signal of p53 and prevents nuclear translocation [56]. Upon DNA damage, p53 is activated by ATM and ATR by multiple mechanisms to enhance its protein stability and transcription capacity [57]. ATM phosphorylates MDM2 to compromise its ligase activity and p53 ubiquitination [58,59], while ATR attenuates p53 nuclear export by phosphorylation of MDM2 at S407 [60]. Collectively, this ATR/ATM–MDM2/MDMX–p53 axis is essential for the DNA damage-induced p53 stabilization and transcriptional control to trigger cell-cycle arrest, DNA repair, and apoptosis events as part of canonical DDR phenotypic responses. 

## 3. Noncanonical Stimuli of DDR

While DDR was first identified in the cellular responses to various types of DNA damage, recent studies have revealed that DDR may also be activated by many noncanonical stimuli [61]. These noncanonical stimuli include viral infection, mechanical stimuli, metabolic cues, and oxidative stresses. 

Viral infection has been shown to activate DDR by incoming viral DNA during the integration of retroviruses or in response to aberrant DNA structures during DNA replication. Therefore, the attenuation of DDR by viruses is an essential aspect of the host evasion and oncogenesis of the virally infected cells. DDR can also be triggered by various mechanical forces generated during chromosome rearrangement and condensation to cope with the resulting mechanical strains [62]. Interestingly, Hippo pathways are responsible for sensing and transmitting the mechanical forces of mammalian cells [63]. Multiple components of the Hippo pathways have been shown to activate DDR [64,65,66,67]. Most relevantly, various forms of oxidative stress also trigger DDR. For example, the oxidation of ATM can trigger the dimerization and activation of ATM in the absence of any DNA damage [68]. The oxidative stress-induced ATM is critical for the chemokine induction and survival of oxidative stresses [69,70]. 

Collectively, these studies show that various components of DDR are modular. Whether functioning as a DNA damage sensor, a signal transducer, or a downstream effector, those DDR components may be decoupled and combined with other physiological processes in a distinct manner under various biological processes. This concept is further enhanced by the role of DDR pathways in the regulation of ferroptosis.

## 4. The Involvement of Various DDR Components in Ferroptosis

### 4.1. ATM and ATR

To identify the role of kinases in ferroptosis, we performed a forward genetic screen to identify kinases essential for cell death triggered by cystine deprivation. Unexpectedly, both ATM and ATR emerged as the top hits, with a strong connection to other kinase hits based on predicted interaction networks. The silencing of ATM or ATR offered robust ferroptosis protection phenotypes, thus validating their essential roles in ferroptosis [71]. Moreover, different ATM inhibitors also rescued multiple cells from ferroptosis-inducing conditions. However, we found that canonical ATM targets, such as Chk2 and p53, were not relevant to the ferroptosis protection associated with ATM inhibition. Instead, ATM inhibition significantly induced the expression of multiple iron regulators involved in iron sequestration (ferritin) and export (ferroportin, *FPN1*). As expected, we found these coordinated changes reduced the levels of labile iron, thus preventing iron-dependent ferroptosis. 

Furthermore, the induction of these iron regulators upon ATM inhibition is mediated by the nuclear translocation of metal regulatory transcription factor 1 (MTF1). Together, the noncanonical function of ATM regulates iron metabolism and ferroptosis sensitivity through MTF1. It is also important to note that we performed our experiments under normal conditions without ionizing radiation or any agents which can induce DNA damage. Moreover, we could not find any evidence that erastin or cystine deprivation can induce DNA damage (phosphorylation of H2AX) or enhance ATM phosphorylation at Ser1981. Therefore, in our observation, a baseline activation of ATM is responsible for the maintenance of labile iron and ferroptosis sensitivity.

### 4.2. Ionizing Radiation and Ferroptosis

One key implication of our study is the potential interaction between IR and ferroptosis. Since IR induces DNA damage and activates ATM [71], it is reasonable to speculate that IR therapy may enhance ferroptosis. Indeed, the Yu and Chen groups showed that IR could further sensitize the ferroptosis of nonsmall cell lung carcinoma (NSCLC) in a GPX4-dependent manner [72]. In another study, Shibata and colleagues also demonstrated the synergistic antitumor effect of erastin and X-ray irradiation through the delayed growth of xenograft models [73].

How does IR sensitize cancer cells to ferroptosis? Three recent studies aimed to elucidate the underlying mechanism. In one study by Zhou’s group, they demonstrated that X-ray irradiation increased lipid ROS, while the ferroptosis antagonist, liproxstatin-1, rescued the amount of lipid ROS. X-ray irradiation also led to the depletion of intracellular glutathione, reduction of *SLC7A11* mRNA, and activation of the ATM pathway. Interestingly, ATM inhibition by Ku-60019 increased the expression of *SLC7A11* under IR, connecting ATM to the glutathione metabolism upon IR [74]. Stockwell’s group also reported similar IR-mediated ferroptosis through enhancing lipid peroxidation and reducing glutathione. Consistent with our findings, there was no correlation between H2AX phosphorylation and ferroptosis. Instead, the relevant ferroptosis determinants that synergize with IR were localized in the cytosol [75]. Therefore, their data indicate that IR can trigger ferroptosis without the involvement of H2AX phosphorylation. Another study by Gan and colleagues also revealed similar interactions between DNA damage response and ferroptosis. They demonstrated that cell death induced by IR could be mitigated by necrosis, apoptosis, ferroptosis inhibitors, and ROS scavengers.

Furthermore, IR induced the expression of many ferroptosis regulators (*GPX4*, *SLC7A11*, and *ACSL4*), and ACSL4 induction contributed significantly to IR-induced ferroptosis [76]. Of note, the deficiency of KEAP1, a negative regulator of NRF2, showed inhibition of IR-ferroptosis as well. This finding is consistent with the consensus knowledge regarding the KEAP1–NRF2 pathway, which is known to robustly and rapidly regulate oxidative stress and ferroptosis [10,11,77]. 

Taken together, IR induces lipid peroxidation and ferroptosis independent of the canonical DDR pathway.

### 4.3. p53 and Ferroptosis

P53 is the gatekeeper of the DNA genome and DNA stability. Upon DNA damage, p53 is stabilized by dissociation from MDM2 and triggers growth arrest, stress response, or programmed cell death to prevent the proliferation of cancerous cells. While p53 is recognized to regulate apoptosis and necrosis, recent studies also implicate p53 in the regulation of ferroptosis. However, the effect of p53 in ferroptosis may vary based on the p53 variants and biological contexts. Here, we also summarize the studies that have shown the promotion and limitation of ferroptosis by p53.

### 4.4. The Promotion of Ferroptosis by p53

Gu and colleagues were the first to report that p53 promotes ferroptosis through repressing *SLC7A11* mRNA by directly occupying the regulatory regions of the *SLC7A11* locus [78]. Consequentially, NRF2 is the canonical transactivator for *SLC7A11*. It is also intriguing that mutant p53 can repress SLC7A11 expression by trapping NRF2 [79], which provides another explanation for the ferroptosis promotion effects of p53. However, it is not clear whether wild-type p53 also shows a similar property to trap NRF2. In a separate study, p53 promoted ferroptosis through the repression of *SLC7A11* mRNA via the H2Bub1-mediated epigenetic mechanism [80]. In two follow-up studies [81,82], Gu’s group also identified two additional p53-dependent regulators for ferroptosis. First, p53 induced the expression of SAT1 (spermidine/spermine *N1*-acetyltransferase 1), the rate-limiting enzyme for polyamine catabolism. Interestingly, SAT1 triggers ferroptosis by enhancing lipid peroxidation in response to ROS [81]. In a subsequent study, they identified ALOX12 (arachidonate 12-lipooxygenase) as a novel positive regulator of ferroptosis [82]. The depletion of ALOX12 robustly abolishes erastin, RSL3, or p53-mediated ferroptosis. Mechanistically, the authors also demonstrated that SLC7A11 binds to ALOX12 and represses its activity. Interestingly, the tumor-derived missense ALOX12 mutants lost their lipoxygenase activity and ferroptosis-inducing capacity, suggesting a potential role of inadequate ferroptosis-mediated tumor suppression [82]. 

The role of p53 in positively regulating ferroptosis is further supported by the identification of an African-centric single nucleotide polymorphism (SNP) on codon 47 (Pro47Ser, rs1800371, G/A) of p53. The P47S p53 variant cells, similar to p53-/- MEFs, are insensitive to ferroptosis and maintain a higher GSH/GSSG ratio and coenzyme A level [83]. Noteworthily, these ferroptotic defects cause iron accumulation and lead toward the differentiation of anti-inflammatory macrophages. The altered immune profiles allow more productive protection of intracellular bacterial infections against malarial toxin hemozoin [84]. Therefore, ferroptosis also plays a role in the elimination of iron-accumulated macrophages and the shaping of host immune responses [84].

Collectively, these studies have shown that p53 promotes ferroptosis via multiple mechanisms. These include the upregulation of *SAT1* or repression of *SLC7A11*, thereby relieving the ferroptosis suppression function of ALOX12 [82]. It would be interesting to investigate the potential connection between p53’s regulation of ferroptosis in the context of ATM/ATR and IR.

### 4.5. The Restriction of Ferroptosis by p53

In contrast to the previous studies, p53 has been reported to inhibit ferroptosis in specific cancer types. Tang’s and Kroemer’s groups first showed that p53 inhibited ferroptosis in colorectal cancer (CRC) [85] through the regulation of subcellular localization of DPP4 (dipeptidyl-peptidase-4). In the p53-null CRC, DPP4 is at the plasma membrane together with NOX1 to enhance lipid peroxidation and ferroptosis. The restoration of wild-type p53 in these cells triggers the nuclear translocation of DPP4, NOX1 dissociation, to restrict lipid peroxidation. In another study from Dixon’s group, the forced expression of wild-type p53 and the treatment of the MDM2 inhibitor (nutlin-3) consistently suppressed ferroptosis through the induction of a canonical p53 target gene, p21 [86]. Mechanistically, the p53-induced p21 upregulation promotes the import of cystine and GSH accumulation to protect cells from ferroptosis. Therefore, the restoration and activation of wild-type p53 may also protect cancer cells from ferroptosis.

### 4.6. MDM2–MDMX: p53-Independent Ferroptosis

MDM2–MDMX are the primary negative regulators of p53 [49]. Interestingly, a recent study has suggested that MDM2–MDMX regulates ferroptosis in a p53-independent manner. MDM2 antagonists (nutlin-3 or MEL23) and MDMX inhibitors both protect cells against ferroptosis by increasing FSP1 proteins and levels of coenzyme Q10 [87]. Furthermore, MDM2–MDMX promotes lipid peroxidation and ferroptosis through lipid reprogramming via PPARα (peroxisome, proliferator-activated receptor α) [49].

### 4.7. Regulation of Ferroptosis via Noncanonical DDR Mechanisms

The canonical DDR pathway is activated by DNA damage to trigger proliferation arrest, induce DNA repair mechanisms, and, when appropriate, execute apoptotic death to remove the cells with unresolved DNA damage. While many DDR components regulate ferroptosis (Figure 1), most regulations occur via noncanonical mechanisms. For example, the activated ATM affects ferroptosis through regulating MTF1 and iron metabolisms instead of its canonical targets, p53 and Chk2. Ionizing radiation induces ferroptosis through the repression of xCT and cystine import (Figure 2). Similarly, p53 regulates ferroptosis by affecting multiple mechanisms, including the expression of *SLC7A11*, *ACSL4*, and *ALOX12* as well as the translocation of DPP4. Most of these target genes regulating ferroptosis are not directly involved in the canonical phenotypic effects of DDR (proliferation arrest, DNA repair, or apoptosis). MDM2/MDMX affects ferroptosis through the induction of FSP1 and the increase of CoQ_10_, but not through their canonical function of regulating p53. Collectively, most components in the DDR pathways affect ferroptosis using noncanonical mechanisms. Therefore, it is tempting to speculate that ferroptosis may be considered a back-up death mechanism of canonical apoptotic cell death for cells with unresolved DNA damage. Another potential but seemingly direct explanation is that the reactive aldehyde products during ferroptosis may eventually trigger DNA damage by reacting with DNA and forming adducts [88]. While most studies did not observe canonical DNA damage by ferroptosis-inducing agents, chronic exposure to ferroptosis-inducing conditions may still lead to the accumulation of DNA damage, which in turn triggers canonical DDR. 

## 5. Therapeutic Implications

### 5.1. The Potential of Ferroptosis to Enhance the Efficacy of Radiotherapies

IR is a standard therapy for many tumors. ATM and ATR are activated during radiation to sense and repair DNA damage caused by ionizing radiation. Moreover, the cell death induced by IR depends on the apoptosis mediated by p53 activation. However, the efficacy of IR can be limited by somatic mutations and microenvironmental factors [89,90], such as hypoxia [91] and acidosis [92]. Therefore, there is significant interest in identifying methods to mitigate radioresistance and enhance the efficacy of ionizing radiation. Thus, the intersection between ferroptosis and DDR suggests that inducing ferroptosis may overcome radioresistance and improve the response (Figure 2). This concept has been supported by several studies that have shown synergistic effects between IR and ferroptosis in various tumor models mentioned previously [72,73,75,76]. As an extension of this concept, it is possible that other cancer therapeutics that trigger DNA damage responses, such as PARP inhibitors or cisplatin, may synergize with ferroptosis-inducing agents for maximal clinical benefits. Furthermore, in patients who are at high risk for developing cancers because of a deficiency in the Fanconi anemia/BRCA/DNA damage response pathway, DNA damage may accumulate in these premalignant cells during oncogenesis. Current guidelines for cancer prevention in BRCA1 mutation carriers may include prophylactic surgery or annual screening with mammography and MRI. These premalignant cells, with significant DNA damage and activated DDR, may be sensitized to ferroptosis. Therefore, inducing ferroptosis in these high-risk patients may eliminate these premalignant cells as a novel prevention strategy.

### 5.2. Repurpose ATM, ATR, and MDM2 Inhibitors to Treat Ferroptosis-Associated Diseases

Other than cancers, ferroptosis has been recognized to be highly relevant to cell death and the pathogenesis of human pathological conditions. For example, various neurodegenerative diseases such as Huntington disease (HD), Alzheimer’s disease (AD), Parkinson’s disease (PD), and amyotrophic lateral sclerosis (ALS) lack curative treatments. Interestingly, the pathogenesis of these diseases involves protein aggregation, excessive ROS, and iron accumulation. Iron accumulation promotes protein aggregation and causes neurodegenerative diseases to directly produce ROS [93,94,95,96]. Increased ROS is known to damage mitochondria, creating a feed-forward loop and increasing ROS production [97], which, in turn, further drives protein aggregation [98].

Furthermore, ROS and protein aggregation can trigger neuroinflammation [99,100]. Neuroinflammation may trigger the release of proinflammatory cytokines that stimulate iron uptake by neurons [101]. This overload of iron accumulation may provide sufficient free iron to drive ferroptosis. Therefore, these positive feedback loops may create an environment that promotes ferroptosis and neuronal death. Consistently, several recent studies have implicated ferroptosis as a critical driver of neurodegeneration [102,103]. Moreover, ferroptosis has shown to be involved in the pathogenesis of other human diseases, including glutamate-induced neuronal death [3], kidney injury [104], and ischemia-reperfusion injuries [105], liver fibrosis [106], cardiomyopathies [107], and heart failure [107]. These diseases typically have features of excessive oxidative stresses, iron accumulation, or lipid peroxidation. In these pathological conditions, ferroptosis may be responsible for the resulting cell death and tissue damage. Therefore, various ferroptosis inhibitors may significantly reduce cell death, delay disease progression, and improve clinical outcomes. While canonical ferroptosis inhibitors show promise in different preclinical models, their translation potential is still unknown.

In contrast, many components in the DDR pathways, such as ATM, ATR, and MDM2/MDMX, are promising therapeutic targets in cancer biology. Therefore, many potent and specific inhibitors have been under development for treating human cancers. For example, multiple ATM inhibitors have been identified as radio-sensitizers to enhance the response to IR. Currently, three ATM inhibitors (AZD0156, KU-60019, AZD1390) are being evaluated in clinical trials for solid tumors [108]. Targeting ATR is also under intensive development, with four potent and selective ATR inhibitors (M6620, M4344, AZD6738, and BAY1895344) in clinical development, either as monotherapy or in combination. Similarly, since the discovery of nutlins (the first MDM2 inhibitor), several compounds have been developed to target MDM2/MDMX to enhance the levels and activities of p53 for antitumor effects [109]. One of the MDM2 inhibitors, AMG-232, has been included in NCI (National Cancer Institute) clinical trials for different solid and liquid tumors [110].

With the new insight that DDR is critical to ferroptosis, these existing inhibitors of DDR may offer therapeutic gain. For example, the inhibitors of ATM, ATR, and MDM2 may protect neuronal death and improve outcomes in neurodegenerative and other ferroptosis-associated diseases that have no effective curative treatments. Similarly, the inhibitors of ATM, ATR, and MDM2 may also have significant therapeutic potential for various ferroptosis-associated human diseases by blocking ferroptotic cell death.

## 6. Conclusions

In summary, many components of DDR regulate ferroptosis by affecting the function of GPX4 and FSP1 and their respective cofactors, GSH and CoQ10. ATM and p53 also regulate ferroptosis by affecting iron and lipid metabolisms, respectively. Interestingly, most of the DDR components affect ferroptosis through noncanonical mechanisms. Therefore, the DDR pathway is highly modular, with distinct sensing mechanisms, signaling transmission, and final effectors. The crosstalk between DDR and ferroptosis further illustrates the highly versatile and diverse outcomes of the different modules of DDR pathways. Such interactions between DDR and ferroptosis provide a strong rationale and mechanistic basis to combine ionizing radiation and ferroptosis-inducers for synergistic effects. Furthermore, various ATM/ATR and MDM2 inhibitors may protect ferroptosis and be used to prevent cell death, delay disease progression, and improve clinical outcomes in many ferroptosis-related human diseases.

## Figures and Tables

**Figure 1 biology-09-00187-f001:**
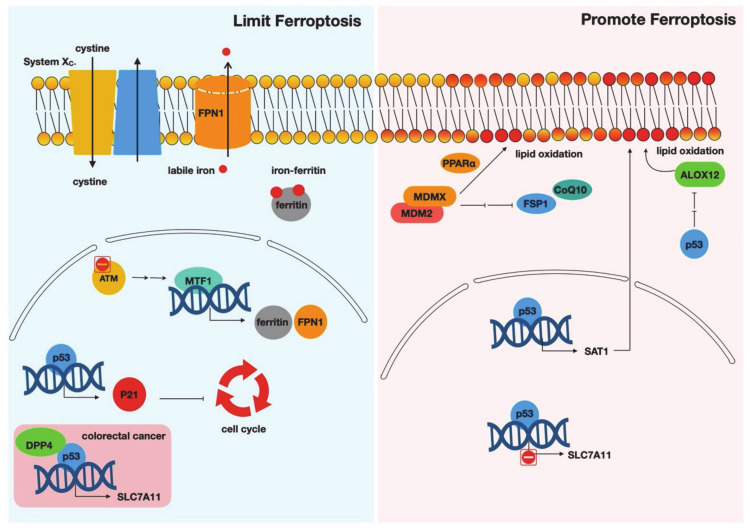
Canonical DNA damage response (DDR) components in ferroptosis. ATM (ataxia–telangiectasia mutated)–MTF1 (metal regulatory transcription factor 1), p53–p21, or p53–DPP4 (dipeptidyl-peptidase-4) axes limit ferroptosis whereas p53–SAT1 (spermidine/spermine N1-acetyltransferase 1), p53–ALOX12 (arachidonate 12-lipooxygenase), or MDM2 (mouse double minute 2)/MDMX (murine double minute X) axes promote ferroptosis.

**Figure 2 biology-09-00187-f002:**
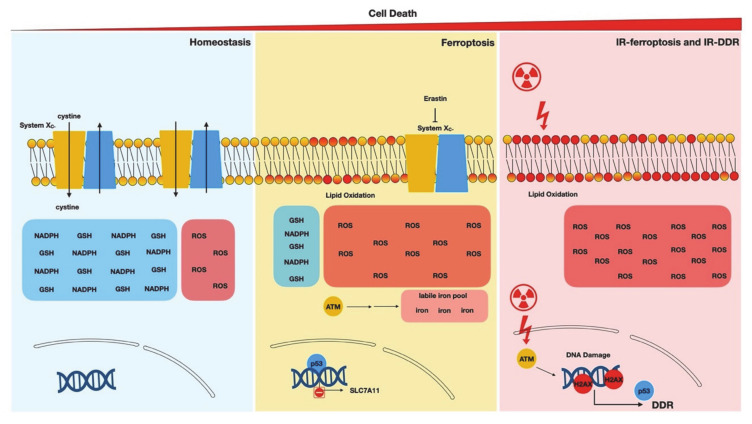
Ionizing radiation (IR) and DDR disrupt ferroptosis protection mechanisms. Imbalanced glutathione (GSH), NADPH, ROS (reactive oxygen species), labile iron, and lipid peroxidation are critical signatures of ferroptosis. Ionizing radiation (IR) increases ROS, lipid peroxidation, and stimulates canonical DDR to eradicate tumor cells synergistically.

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
