# Peer review of "The Intersection of DNA Damage Response and Ferroptosis—A Rationale for Combination Therapeutics"

_biology, 2020, doi:10.3390/biology9080187_

Round 1
Reviewer 1 Report
The authors reviewed recent findings on interactions between ferroptosis and DNA damage response pathway with an emphasis on ionization radiation induced damage, which has clinical implications.
Is ferroptosis cell type specific?
NADPH in Figure 2?
Line 37: may want to spell out glutathione for GSH for the first time
Ref 8 and 9: may want to update the citation
Author Response
The authors reviewed recent findings on interactions between ferroptosis and DNA damage response pathway with an emphasis on ionization radiation induced damage, which has clinical implications.
Is ferroptosis cell type specific?
Response:
There is some cell-type specificity for ferroptosis. For example, a survey of many cancer cells showed that Diffuse Large B Cell Lymphomas and Renal Cell Carcinomas are highly sensitive to ferroptosis (1). However, ferroptosis susceptibility has been described for a broad range of cancer cells from different tissues, including renal, breast, pancreatic, lymphoma and other tumor cells. We have cited many of these studies in our review.
NADPH in Figure 2?
Response:
Yes, we have modified the figure 2 and incorporate the NADPH into the Figure 2.
Line 37: may want to spell out glutathione for GSH for the first time
Response:
Thank you for your suggestions. We have spelt out glutathione for GSH for the first time.
Ref 8 and 9: may want to update the citation
Response:
Thank you for your suggestions. We have updated the citations for these two references.
Reviewer 2 Report
In this manuscript by Chen et al, the authors present a timely and interesting review paper about the intersection of DNA damage response (DDR) and ferroptosis with discussion of its clinical application. The crosstalk between DDR and ferroptosis, as a new form of program cell death, are of general interest to the field. However, there are several concerns that need to be addressed before this manuscript could be published.
First, the general writing, grammar and sentence structure need improvement. There are many common grammatical errors like subject-verb disagreement (too many to point out), frequent wrong word usages (e.g. using "genetical" instead of "genetic"; "It is worth to note" instead of "noting"), and some spelling errors (e.g. "ubiquination", "Huntingdon Disease" etc.).
Second, since the authors claimed that the crosstalk between DDR and ferroptosis are mediated mostly via non-canonical mechanisms, it would help readers better understand the regulatory pathway by adding the background introduction about non-canonical mechanism of DDR and reducing the part of canonical mechanism accordingly.
Finally, it would be better for the authors to follow the flow by reorganization of section 3, especially for the part of p53 and ferroptosis.
Minor concern:
- The authors should avoid repeat definition of abbreviation like "DSB".
- The authors may consider adding a section of conclusions.
Author Response
Comments and Suggestions for Authors
In this manuscript by Chen et al, the authors present a timely and interesting review paper about the intersection of DNA damage response (DDR) and ferroptosis with discussion of its clinical application. The crosstalk between DDR and ferroptosis, as a new form of program cell death, are of general interest to the field. However, there are several concerns that need to be addressed before this manuscript could be published.
First, the general writing, grammar and sentence structure need improvement. There are many common grammatical errors like subject-verb disagreement (too many to point out), frequent wrong word usages (e.g. using "genetical" instead of "genetic"; "It is worth to note" instead of "noting"), and some spelling errors (e.g. "ubiquination", "Huntingdon Disease" etc.).
Response:
Thank you for your suggestions. We have gone through the manuscript carefully and improved spelling and grammar.
Second, since the authors claimed that the crosstalk between DDR and ferroptosis are mediated mostly via non-canonical mechanisms, it would help readers better understand the regulatory pathway by adding the background introduction about non-canonical mechanism of DDR and reducing the part of canonical mechanism accordingly.
Response:
Thank you for your suggestions. We have now included a new section on the non-canonical mechanism of DDR in the revised manuscript. However, the extensive review of the non-canonical function beyond ferroptosis is definitely beyond the scope of the current review.
Finally, it would be better for the authors to follow the flow by reorganization of section 3, especially for the part of p53 and ferroptosis.
Response:
Thank you for your suggestions. We have slightly modified the relevant sections.
Minor concern:
- The authors should avoid repeat definition of abbreviation like "DSB".
- The authors may consider adding a section of conclusions.
Response:
Thank you for your suggestions. We have made the suggested changes and avoided the repeated definition of DSB. In addition, we have added a section of the conclusion at the end of the review article.
Reviewer 3 Report
No comment. The review is well organized.
Author Response
No comment. The review is well organized.
Response:
Thank you for your nice comments.
Reviewer 4 Report
This review recapitulates the thesis put forward by the authors in their original research articles, that described the importance of ferroptosis in cancer and liver fibrosis. This article is informative because the authors tried presenting available information on ferroptosis and discussed its association with DNA damage response. In my opinion, this could be of great interest to illuminate this topic and will be a considerable contribution in the field, if, in general, authors could take care of manuscript organization.
Please find some concerns below:
1. I appreciate the sincere efforts of the authors to summarize the available information on the given subject, using graphics, however, it would be better if authors can re-organize this manuscript in a proper structured way. The authors need to work on connection between different paragraphs/sections throughout the manuscript. My specific concern is with introductory part “Ferroptosis-biological processes and genetic determinants.” This section is highly fragmented, and as it is a starter, it will distract readers to read further.
2. Line 31-33, “we have made significant progress…”missing references which can support this statement.
3. 34-35, Most mammalian cells….ferroptosis” all of sudden lipid peroxidation appears and then they jump to glutathione peroxidase, without any mention how lipid peroxidation is associated with ferroptosis. They discussed relation between lipid peroxidation and ferroptosis in paragraph sixth, line 61-66.
4. In paragraph third, they discussed cysteine and its association with GSH, but ended with role of cysteine in cancer, without any mention of role of ferroptosis in cancer.
5. Last paragraph of the first section, ended with high emphasis on ferroptosis in cancer, while reading abstract and title of manuscript it appears that authors going to talk about association of ferroptosis with DDR and may link that with cancer. Therefore, it is recommended to rewrite first section of manuscript.
6. The authors used a lot of abbreviations in the manuscript and didn’t provide elaboration of these terms. It is advisable to elaborate them as possible.
Author Response
This review recapitulates the thesis put forward by the authors in their original research articles, that described the importance of ferroptosis in cancer and liver fibrosis. This article is informative because the authors tried presenting available information on ferroptosis and discussed its association with DNA damage response. In my opinion, this could be of great interest to illuminate this topic and will be a considerable contribution in the field, if, in general, authors could take care of manuscript organization.
Response:
Thank you for your nice comments.
Please find some concerns below:
1. I appreciate the sincere efforts of the authors to summarize the available information on the given subject, using graphics, however, it would be better if authors can re-organize this manuscript in a proper structured way. The authors need to work on connection between different paragraphs/sections throughout the manuscript. My specific concern is with introductory part “Ferroptosis-biological processes and genetic determinants.” This section is highly fragmented, and as it is a starter, it will distract readers to read further.
Response:
Thank you for your suggestions. We have re-organized the introductory part on the ferroptosis background into several pathways to facilitate the discussion of these processes with DDR in the later section.
Line 31-33, “we have made significant progress…” missing references which can support this statement.
Response:
Thank you for your suggestions. We have added some ferroptosis review articles to support this statement.
34-35, Most mammalian cells….ferroptosis” all of sudden lipid peroxidation appears and then they jump to glutathione peroxidase, without any mention how lipid peroxidation is associated with ferroptosis. They discussed relation between lipid peroxidation and ferroptosis in paragraph sixth, line 61-66.
Response:
Thank you for your suggestions. We have re-arranged the texts and made suggested changes to have better transitions.
In paragraph third, they discussed cysteine and its association with GSH, but ended with role of cysteine in cancer, without any mention of role of ferroptosis in cancer.
Response:
Thank you for your suggestions. We will make the suggested changes in the particular sections.
Last paragraph of the first section, ended with high emphasis on ferroptosis in cancer, while reading abstract and title of manuscript it appears that authors going to talk about association of ferroptosis with DDR and may link that with cancer. Therefore, it is recommended to rewrite first section of manuscript.
Response:
Thank you for your suggestions. We will re-write the first section of the manuscript.
The authors used a lot of abbreviations in the manuscript and didn’t provide elaboration of these terms. It is advisable to elaborate on them as possible.
Response:
Thank you for your suggestions. We have elaborated on these abbreviations in the revised manuscript.